# Learning Certified Individually Fair Representations

**Anian Ruoss, Mislav Balunović, Marc Fischer, Martin Vechev**
Department of Computer Science
ETH Zurich
anruoss@ethz.ch
{mislav.balunovic, marc.fischer, martin.vechev}@inf.ethz.ch

## Abstract

Fair representation learning provides an effective way of enforcing fairness constraints without compromising utility for downstream users. A desirable family of such fairness constraints, each requiring similar treatment for similar individuals, is known as individual fairness. In this work, we introduce the first method that enables data consumers to obtain certificates of individual fairness for existing and new data points. The key idea is to map similar individuals to close latent representations and leverage this latent proximity to certify individual fairness. That is, our method enables the data producer to learn and certify a representation where for a data point all similar individuals are at $\ell_\infty$-distance at most $\epsilon$, thus allowing data consumers to certify individual fairness by proving $\epsilon$-robustness of their classifier. Our experimental evaluation on five real-world datasets and several fairness constraints demonstrates the expressivity and scalability of our approach.

## 1 Introduction

The increased use of machine learning in sensitive domains (e.g., crime risk assessment [1], ad targeting [2], and credit scoring [3]) has raised concerns that methods learning from data can reinforce human bias, discriminate, and lack fairness [4–6]. Moreover, data owners often face the challenge that their data will be used in (unknown) downstream applications, potentially indifferent to fairness concerns [7]. To address this challenge, the paradigm of learning fair representations has emerged as a promising approach to obtain data representations that preserve fairness while maintaining utility for a variety of downstream tasks [8, 9]. The recent work of McNamara et al. [10] has formalized this setting by partitioning the landscape into: a *data regulator* who defines fairness for the particular task at hand, a *data producer* who processes sensitive user data and transforms it into another representation, and a *data consumer* who performs predictions based on the new representation.

In this setting, a machine learning model $M \colon \mathbb{R}^n \to \mathbb{R}^o$ is composed of two parts: an encoder $f_\theta \colon \mathbb{R}^n \to \mathbb{R}^k$, provided by the data producer, and a classifier $h_\psi \colon \mathbb{R}^k \to \mathbb{R}^o$, provided by the data consumer, with $\mathbb{R}^k$ denoting the latent space. The data regulator selects a definition of fairness that the model $M$ should satisfy. Most work so far has explored two main families of fairness definitions [11]: *statistical* and *individual*. Statistical notions define specific groups in the population and require that particular statistics, computed based on model decisions, should be equal for all groups. Popular notions of this kind include demographic parity [12] and equalized odds [13]. While these notions do not require any assumptions on the data and are easy to certify, they offer no guarantees for individuals or other subgroups in the population [14]. In contrast, individual notions of fairness [12] are desirable as they explicitly require that similar individuals in the population are treated similarly.

**Key challenge**    A central challenge then is to enforce individual fairness in the setting described above. That is, to both learn an individually fair representation and to certify that individual fairness

is actually satisfied across the end-to-end model $M$ without compromising the independence of the data producer and the data consumer.

**This work**  In this work, we propose the first method for addressing the above challenge. At a high level, our approach is based on the observation that recent advances in training machine learning models with logical constraints [15] together with new methods for proving that constraints are satisfied [16] open the possibility for learning certified individually fair models.

Concretely, we identify a practical class of individual fairness definitions captured via declarative fairness constraints. Such a fairness constraint is a binary similarity function $\phi \colon \mathbb{R}^n \times \mathbb{R}^n \to \{0, 1\}$, where $\phi(x, x')$ evaluates to 1 if and only if two individuals $x$ and $x'$ are similar (e.g., if all their attributes except for race are the same). By working with declarative constraints, data regulators can now express interpretable, domain-specific notions of similarity, a problem known to be challenging [8, 17–21].

Given the fairness constraint $\phi$, we can now train an individually fair representation and use it to obtain a certificate of individual fairness for the end-to-end model. For training, the data producer can employ our framework to learn an encoder $f_\theta$ with the goal that two individuals satisfying $\phi$ should be mapped close together in $\ell_\infty$-distance in latent space. As a consequence, individual fairness can then be certified for a data point in two steps: first, the data producer computes a convex relaxation of the latent set of similar individuals and passes it to the data consumer. Second, the data consumer certifies individual fairness by proving local robustness within the convex relaxation. Importantly, the data consumer can now perform *modular* certification: it does not need to know the fairness constraint $\phi$ and the concrete data point $x$.

Our experimental evaluation on several datasets and fairness constraints shows a substantial increase (up to 72.6%) of certified individuals (unseen during training) when compared to standard representation learning.

**Main contributions**  Our key contributions are:

- A practical family of similarity notions for individual fairness defined via interpretable logical constraints.

- A method to learn individually fair representations (defined in an expressive logical fragment), which comes with provable certificates.

- An end-to-end implementation of our method in an open-source tool called LCIFR, together with an extensive evaluation on several datasets, constraints, and architectures. We make LCIFR publicly available at `https://github.com/eth-sri/lcifr`.

## 2  Overview

This section provides a high-level overview of our approach, with the overall flow shown in Figure 1.

As introduced earlier, our setting consists of three parties. The first party is a data regulator who defines similarity measures for the input and the output denoted as $\phi$ and $\mu$, respectively. The properties $\phi$ and $\mu$ are problem-specific and can be expressed in a rich logical fragment which we describe later in Section 4. For example, for classification tasks $\mu$ could denote equal classification (i.e., $\mu(M(x), M(x')) = 1 \iff M(x) = M(x')$) or classifying $M(x)$ and $M(x')$ to the same label group; for regressions tasks $\mu$ could evaluate to 1 if $\|M(x) - M(x')\| \leq 0.1$ and 0 otherwise. We focus on equal classification in the classification setting for the remainder of this work.

The goal of treating similar individuals as similarly as possible can then be formulated as finding a classifier $M$ which maximizes

$$\mathbb{E}_{x \sim D}\left[\forall x' \in \mathbb{R}^n : \phi(x, x') \implies \mu(M(x), M(x'))\right], \tag{1}$$

where $D$ is the underlying data distribution (we assume a logical expression evaluates to 1 if it is true and to 0 otherwise). As usual in machine learning, we approximate this quantity with the empirical risk, by computing the percentage of individuals $x$ from the test set for which we can certify that

$$\forall x' \in S_\phi(x) : \mu(M(x), M(x')), \tag{2}$$

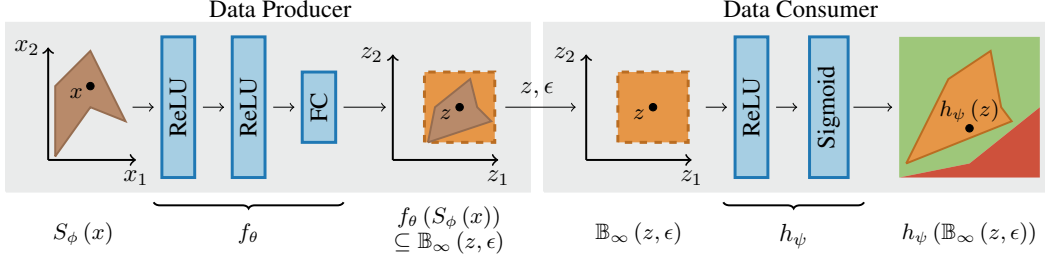

Figure 1: Conceptual overview of our framework. The left side shows the component corresponding to the data producer who learns an encoder $f_\theta$ which maps the entire set of individuals $S_\phi(x)$ that are similar to individual $x$, according to the similarity notion $\phi$, to points near $f_\theta(x)$ in the latent space. The data producer then computes an $\ell_\infty$-bounding box $\mathbb{B}_\infty$ around the latent set of similar individuals $f_\theta(S_\phi(x))$ with center $z = f_\theta(x)$ and radius $\epsilon$ and passes it to the data consumer. The data consumer receives the latent representation $z$ and radius $\epsilon$, trains a classifier $h_\psi$, and certifies that the entire $\ell_\infty$-ball centered around $z$ with radius $\epsilon$ is classified the same (green color shows fair output region).

where $S_\phi(x) = \{x' \in \mathbb{R}^n \mid \phi(x, x')\}$ denotes the set of all points similar to $x$. Note that $S_\phi(x)$ generally contains an infinite number of individuals. In Figure 1, $S_\phi(x)$ is represented as a brown shape, and $x$ is shown as a single point inside of $S_\phi(x)$.

The key idea of our approach is to train the encoder $f_\theta$ to map point $x$ and all points $x' \in S_\phi(x)$ close to one another in the latent space with respect to $\ell_\infty$-distance, specified as

$$\phi(x, x') \implies ||f_\theta(x') - f_\theta(x)||_\infty \leq \delta, \tag{3}$$

where $\delta$ is a tunable parameter of the method, determined in agreement between producer and consumer (we could also use another $\ell_p$-norm). If the encoder indeed satisfies Equation (3), the data consumer, potentially indifferent to the fairness constraint, can then train a classifier $h_\psi$ independently of the similarity notion $\phi$. Namely, the data consumer only has to train $h_\psi$ to be robust to perturbations up to $\delta$ in $\ell_\infty$-norm, which can be solved via standard min-max optimization, discussed in Section 4.

We now explain our end-to-end inference with provable certificates for encoder $f_\theta$ and classifier $h_\psi$.

**Processing the producer model**   Given a data point $x$, we first propagate both $x$ and its set of similar points $S_\phi(x)$ through the encoder, as shown in Figure 1, to obtain the latent representations $z = f_\theta(x)$ and $f_\theta(S_\phi(x))$. As Equation (3) may not hold for the particular $x$ and $\delta$ due to the stochastic nature of training, we compute the smallest $\ell_\infty$-bounding box of radius $\epsilon$ such that $f_\theta(S_\phi(x)) \subseteq \mathbb{B}_\infty(z, \epsilon) := \{z' \mid ||z - z'||_\infty \leq \epsilon\}$. This $\ell_\infty$-bounding box with center $z$ and radius $\epsilon$ is shown as orange in Figure 1.

**Processing the consumer model**   Next, we provide the latent representation $z$ and the radius $\epsilon$ to the data consumer. The data consumer then knows that all points similar to $x$ are in the $\ell_\infty$-ball of radius $\epsilon$, but does not need to know the similarity constraint $\phi$ nor the particular shape $f_\theta(S_\phi(x))$. The key observation is the following: if the data consumer can prove its classifier $h_\psi$ is robust to $\ell_\infty$-perturbations up to $\epsilon$ around $z$, then the end-to-end classifier $M = h_\psi \circ f_\theta$ satisfies individual fairness at $x$ with respect to the similarity rule $\phi$ imposed by the data regulator.

There are two central technical challenges we need to address. The first challenge is how to train an encoder to satisfy Equation (3), while not making any domain-specific assumptions about the point $x$ or the similarity constraint $\phi$. The second challenge is how to provide a certificate of individual fairness for $x$, which requires both computing the smallest radius $\epsilon$ such that $f_\theta(S_\phi(x)) \subseteq \mathbb{B}_\infty(z, \epsilon)$, as well as certifying $\ell_\infty$-robustness of the classifier $h_\psi$.

To train an encoder, we build on Fischer et al. [15], who provide a translation from logical constraints $\phi$ to a differentiable loss function. The training of the encoder network can then be formulated as a min-max optimization problem, which alternates between (i) searching for counterexamples $x' \in S_\phi(x)$ that violate Equation (3), and (ii) training $f_\theta$ on the counterexamples. We employ gradient descent to minimize a joint objective composed of a classification loss and the constraint loss obtained from translating Equation (3). Once no more counterexamples are found, we can conclude the encoder empirically satisfies Equation (3). We discuss the detailed procedure in Section 4.

We compute a certificate for individual fairness in two steps. First, to provide guarantees on the latent representation generated by encoder $f_\theta$, we solve the optimization problem

$$\epsilon = \max_{x' \in S_\phi(x)} ||z - f_\theta(x')||_\infty.$$

Recall that the set $S_\phi(x)$ generally contains an infinite number of individuals $x'$, and thus this optimization problem cannot be solved by simple enumeration. In Section 5 we show how this optimization problem can be encoded as a mixed-integer linear program (MILP) and solved using off-the-shelf MILP solvers. After obtaining $\epsilon$, we certify local robustness of the classifier $h_\psi$ around $z = f_\theta(x)$ by proving (using MILP) that for each $z'$ where $||z' - z|| \le \epsilon$, the classification results of $h_\psi(z')$ and $h_\psi(z)$ coincide. Altogether, this implies the overall model $M = h_\psi \circ f_\theta$ satisfies individual fairness for $x$. Finally, note that since the bounding box $\mathbb{B}(z, \epsilon)$ is a convex relaxation of the latent set of similar individuals $f_\theta(S_\phi(x))$, the number of individuals for which we can obtain a certificate is generally lower than the number of individuals that actually satisfy Equation (2).

## 3   Related Work

We now discuss work most closely related to ours.

**Learning fair representations**   There has been a long line of work on learning fair representations. Zemel et al. [8] introduced a method to learn fair representations that ensure group fairness and protect sensitive attributes. Such representations, invariant to sensitive attributes, can also be learned using variational autoencoders [22], adversarial learning [9, 23], or disentanglement [24]. Zemel et al. [8] and Madras et al. [9] also consider the problem of fair transfer learning, which we investigate in our work. Song et al. [25] used duality to unify some of the mentioned work under the same framework. McNamara et al. [26] derived theoretical guarantees for learning fair representations. Their guarantees require statistics of the data distribution and consist of probabilistic bounds for individual and group fairness: for a new data point from the same distribution, the constraint will hold with a certain probability. In contrast, we obtain a certificate for a fixed data point, which ensures that the fairness constraints holds (independent of the other data points).

Most work so far focuses on learning representations that satisfy statistical notions of fairness, but there has also been some recent work on learning individually fair representations. These works learn fair representations with alternative definitions of individual fairness based on Wasserstein distance [19, 27], fairness graphs [18], or distance measures [17]. A different line of work has investigated leaning the fairness metric from data [19–21, 28]. In contrast, we define individual fairness via interpretable logical constraints. Finally, recent works [29–31] studied the task of learning representations that are robust to (adversarial) perturbations, i.e., all similar individuals in our case, however not in the context of fairness. Many of the above methods for learning (individually) fair representations employ nonlinear components [8, 17], graphs [18], or sampling [22, 24] and can thus not be efficiently certified, unlike the neural networks that we consider in our work.

While we focus on learning fair representations, other lines of work have investigated individual fairness in the context of clustering [32, 33], causal inference [34–37], composition of individually fair classifiers [38, 39], and differential privacy (DP) [12, 40, 41]. The close relationship between individual fairness and DP has been discussed in previous work (see, e.g., [12]). However, DP crucially differs from our work in that it obtains a probabilistic fairness guarantee, similar to McNamara et al. [26] mentioned above, whereas we compute absolute fairness guarantees for every data point. The most natural way to employ DP for a representation learning approach, like LCIFR, would be to make the data producer model $f_\theta$ differentially private for a neighborhood that encodes $S_\phi$, by adding noise inside the computation of $f_\theta$. If one can achieve DP for the neighborhood $S_\phi$ (a non-trivial challenge), the data consumer model can then be seen as a post-processing step, which with the right robustness certificate yields a probabilistic guarantee of Equation (2).

**Certification of neural networks**   Certification of neural networks has become an effective way to prove that these models are robust to adversarial perturbations. Certification approaches are typically based on SMT solving [42], abstract interpretation [43], mixed-integer linear programming [16], or linear relaxations [44–46]. Three concurrent works also investigate the certification of individual fairness [47–49]. However, these works try to certify a global individual fairness property, i.e., for a given classifier there exists an input which is not treated individually fair, whereas we focus on local

individual fairness, i.e., for every concrete data point we certify whether the model is individually fair or not. Moreover, these works consider similarity notions that are less expressive than and can be captured by our logical constraints. Finally, they only consider certifying fairness of existing models, while we also focus on learning fair representations.

In our work, we investigate modular certification. For the data producer, we need to propagate the input shape through both logical operators (e.g., conjunctions and disjunctions) and the neural network. While in our work, we use a MILP encoding, other approaches could also be applied by crafting specialized convex relaxations. For example, if our approach is applied to learn individually fair representations of complex data such as images, where encoder networks are usually larger than in tabular data that we consider here, one could leverage the certification framework from Singh et al. [46]. On the data consumer side, any of the above approaches could be applied as they are all designed to certify $\ell_\infty$-robustness which we consider in our work.

## 4 Learning Individually Fair Representations

We now present our method for learning individually fair representations with respect to the property $\phi$. To illustrate our method, we consider the case where the regulator proposes the similarity constraint:

$$\phi(x, x') := \bigwedge_{i \in \text{Cat} \setminus \{\text{race}\}} (x_i = x'_i) \bigwedge_{j \in \text{Num}} |x_j - x'_j| \le \alpha.$$

According to $\phi$, individual $x'$ is considered similar to $x$ if: (i) all categorical attributes except for race are equal to those of $x$, and (ii) all numerical attributes (e.g., income) of $x$ and $x'$ differ by at most $\alpha$. Thus, under $\phi$, the similarity of individuals $x$ and $x'$ does not depend on their respective races. Note that since $\phi$ is binary $x$ and $x'$ are either considered similar or not which is in line with the typical use-case in classification where two individuals are either classified to the same label or not. Moreover, such logical formulas (of reasonable size) are generally considered humanly readable and are thus investigated in the interpretable machine learning community (e.g., for decision trees [50]).

**Enforcing individual fairness** To learn a representation that satisfies $\phi$, we build on the recent work DL2 [15]. Concretely, we aim to enforce the following constraint on the encoder $f_\theta$ used by the data producer:

$$\phi(x, x') \implies \|f_\theta(x) - f_\theta(x')\|_\infty \le \delta, \tag{4}$$

where $\delta$ is a tunable constant, determined in agreement between the data producer and the data consumer. With DL2, this implication can be translated into a non-negative, differentiable loss $\mathcal{L}(\phi)$ such that $\mathcal{L}(\phi)(x, x') = 0$ if and only if the implication is satisfied. Here, we denote $\omega(x, x') := \|f_\theta(x) - f_\theta(x')\|_\infty \le \delta$ and translate the constraint in Equation (4) as

$$\mathcal{L}(\phi \implies \omega) = \mathcal{L}(\neg\phi \vee \omega) = \mathcal{L}(\neg\phi) \cdot \mathcal{L}(\omega),$$

where negations are propagated through constraints via standard logic. Moreover, we have

$$\begin{aligned} \mathcal{L}(\omega)(x, x') &= \mathcal{L}(\|f_\theta(x) - f_\theta(x')\|_\infty \le \delta) \\ &= \max\{\|f_\theta(x) - f_\theta(x')\|_\infty - \delta, 0\}. \end{aligned}$$

Similarly, conjunctions $\mathcal{L}(\phi' \wedge \phi'')$ would be translated as $\mathcal{L}(\phi') + \mathcal{L}(\phi'')$, and we refer interested readers to the original work [15] for further details on the translation.

Using this differentiable loss, the data producer can now approximate the problem of finding an encoder $f_\theta$ that maximizes the probability that the constraint $\phi \implies \omega$ is satisfied for all individuals via the following min-max optimization problem (defined in two steps): First, we find a counterexample

$$x^* = \arg\min_{x' \in S_\phi(x)} \mathcal{L}(\neg(\phi \implies \omega))(x, x'),$$

where $S_\phi(x) = \{x' \in \mathbb{R}^n \mid \phi(x, x')\}$ denotes the set of all individuals similar to $x$ according to $\phi$. Then, in the second step, we find the parameters $\theta$ that minimize the constraint loss at $x^*$:

$$\arg\min_\theta \mathbb{E}_{x \sim D}[\mathcal{L}(\phi \implies \omega)(x, x^*)].$$

Note that in the outer loop, we are finding parameters $\theta$ that minimize the loss of the original constraint from Equation (4), while in the inner loop, we are finding a counterexample $x^*$ by minimizing the

loss corresponding to the negation of this constraint. We use Adam [51] for optimizing the outer problem. For the inner minimization problem, Fischer et al. [15] further refine the loss by excluding constraints that have closed-form analytical solutions, e.g., $\max \{\|x - x'\|_\infty - \delta, 0\}$ which can be minimized by projecting $x'$ onto the $\ell_\infty$-ball of radius $\delta$ around $x$. The resulting objective is thus

$$x^* = \arg\min_{x' \in \mathbb{C}} \mathcal{L}(\rho)(x, x'),$$

where $\mathbb{C}$ is the convex set and $\rho$ is $\neg(\phi \implies \omega)$ without the respective constraints. It has been shown [52] that such an objective can be efficiently solved with Projected Gradient Descent (PGD).

DL2 does not provide a meaningful translation for categorical constraints, which are essential to fairness, and we derive a relaxation method for training with categorical constraints in Appendix A.

**Predictive utility of the representation**  Recall that our method is modular in the sense that the data producer and the data consumer models are learned separately. Thus, the data producer needs to ensure that the latent representation remains informative for downstream applications (represented by the data consumer model $h_\psi$). To that end, the data producer additionally trains a classifier $q \colon \mathbb{R}^k \to \mathbb{R}^o$ that tries to predict the target label $y$ from the latent representation $z = f_\theta(x)$. Thus, the data producer seeks to jointly train the encoder $f_\theta$ and classifier $q$ to minimize the combined objective

$$\arg\min_{f_\theta, q} \mathbb{E}_{x,y} \left[ \mathcal{L}_C \left( q \left( f_\theta(x) \right), y \right) + \gamma \mathcal{L}_F \left( x, f_\theta(x) \right) \right], \tag{5}$$

where $\mathcal{L}_C$ is any suitable classification loss (e.g., cross-entropy), $\mathcal{L}_F$ is the fairness constraint loss obtained via DL2, and the hyperparameter $\gamma$ balances the two objectives. We empirically investigate impact of the loss balancing factor $\gamma$ on the accuracy-fairness tradeoff in Appendix B.

**Training robust classifier** $h_\psi$  We assume the encoder $f_\theta$ has been trained to maintain predictive utility and satisfy Equation (4). Recall that, given this assumption, the data consumer who wants to ensure her classifier $h_\psi$ is individually fair, only needs to ensure local robustness of the classifier for perturbations up to $\delta$ in $l_\infty$-norm. This is a standard problem in robust machine learning [53] and can be solved via min-max optimization, recently found to work well for neural network models [52]:

$$\min_\psi \mathbb{E}_{z \sim \mathcal{D}_z} \left[ \max_{\pi \in [\pm\delta]} \mathcal{L}_C \left( h_\psi(z + \pi), y \right) \right],$$

where $D_z$ is the latent distribution obtained by sampling from $\mathcal{D}$ and applying the encoder $f_\theta$, and $\mathcal{L}_C$ is a suitable classification loss. The optimization alternates between: (i) trying to find $\pi \in [\pm\delta]$ that maximizes $\mathcal{L}_C \left( h_\psi(z + \pi), y \right)$, and (ii) updating $\psi$ to minimize $\mathcal{L}_C \left( h_\psi(z + \pi), y \right)$ under such worst-case perturbations $\pi$. While the theoretical necessity of training for local robustness is clear, we empirically investigate its effect on accuracy and certifiable fairness in Appendix C.

## 5  Certifying Individual Fairness

In this section we discuss how the data consumer can compute a certificate of individual fairness for its model $h_\psi$ trained on the latent representation (as described in Section 4 above). We split this process into two steps: (i) the data producer propagates a data point $x$ through the encoder to obtain $z = f_\theta(x)$ and computes the radius $\epsilon$ of the smallest $\ell_\infty$-ball around $z$ that contains the latent representations of all similar individuals $f_\theta(S_\phi(x))$, i.e., $f_\theta(S_\phi(x)) \subseteq \mathbb{B}_\infty(z, \epsilon)$, and (ii) the data consumer checks whether all points in the latent space that differ by at most $\epsilon$ from $z$ are classified to the same label, i.e., $h_\psi(z) = h_\psi(z')$ for all $z' \in \mathbb{B}_\infty(z, \epsilon)$. We now discuss both of these steps.

### 5.1  Certifying Latent Similarity

To compute the minimum $\epsilon$ which ensures that $f_\theta(S_\phi(x)) \subseteq \mathbb{B}_\infty(z, \epsilon)$, the data producer models the set of similar individuals $S_\phi(x)$ and the encoder $f_\theta$ as a mixed-integer linear program (MILP).

**Modeling $S_\phi$ as MILP**  We use an example to demonstrate the encoding of logical constraints with MILP. Consider an individual $x$ that has two categorical features $x_1 = [1, 0, \ldots, 0]$ and $x_2 = [0, \ldots, 0, 1]$ and one numerical feature $x_3$, with the following constraint for similarity:

$$\phi(x, x') := (x_1 = x_1') \wedge (|x_3 - x_3'| \le \alpha).$$

Here $x$ is an individual from the test dataset and can be treated as constant, while $x'$ is encoded using mixed-integer variables. For every categorical feature $x'_i$ we introduce $k$ binary variables $v^l_i$ with $l = 1, \ldots, k$, where $k$ is the number of distinct values this categorical feature can take. For the fixed categorical feature $x'_1$, which is equal to $x_1$, we add the constraints $v^1_1 = 1$ and $v^l_1 = 0$ for $l = 2, \ldots, k$. To model the free categorical feature $x'_2$ we add the constraint $\sum_l v^l_2 = 1$ thereby enforcing it to take on exactly one of $k$ potential values. Finally, the numerical attribute $x'_3$ can be modeled by adding a corresponding variable $v_3$ with the two constraints: $v_3 \geq x_3 - \alpha$ and $v_3 \leq x_3 + \alpha$. It can be easily verified that our encoding of $S_\phi$ is exact.

Consider now a fairness constraint including disjunctions, i.e., $\phi := \phi_1 \vee \phi_2$. To model such a disjunction we introduce two auxiliary binary variables $v_1$ and $v_2$ with the constraints $v_i = 1 \iff \phi_i(x, x') = 1$ for $i = 1, 2$ and $v_1 + v_2 \geq 1$.

**Handling general constraints** The encodings demonstrated on these two examples can be applied for general constraints $\phi$. A full formalization of our encoding is found in Appendix D.

**Modeling $f_\theta$ as MILP** To model the encoder we employ the method from Tjeng et al. [16] which is exact for neural networks with ReLU activations. We recall that a ReLU performs $\max\{x, 0\}$ for some input $x$. Given an upper and lower bound on $x$, i.e., $x \in [l, u]$ we can encode the output of ReLU exactly via case distinction: (i) if $u \leq 0$ add a variable with upper and lower bound $0$ to MILP, (ii) if $l \geq 0$ add a variable with upper and lower bounds $u$ and $l$ respectively to MILP, and (iii) if $l < 0 < u$, add a variable $v$ and a binary indicator $i$ to MILP in addition to the following constraints:

$$0 \leq v \leq x \cdot i,$$
$$x \leq v \leq x - l \cdot (1 - i),$$
$$i = 1 \iff 0 \leq x.$$

Finally, given the MILP formulation of $S_\phi$ and $f_\theta$ we can compute $\epsilon$ by solving the following $k$ MILP instances (where $k$ is the dimension of the latent space):

$$\hat{\epsilon}_j = \max_{x' \in S_\phi(x)} |f^{(j)}_\theta(x) - f^{(j)}_\theta(x')|.$$

We compute the final result as $\epsilon = \max\{\hat{\epsilon}_1, \hat{\epsilon}_2, \ldots \hat{\epsilon}_k\}$.

## 5.2 Certifying Local Robustness

The data consumer obtains a point in latent space $z$ and a radius $\epsilon$. To obtain a fairness certificate, the data consumer certifies that all points in the latent space at $\ell_\infty$-distance at most $\epsilon$ from $z$ are mapped to the same label as $z$. This amounts to solving the following MILP optimization problem for each logit $h^{(y')}_\psi$ with label $y'$ different from the true label $y$:

$$\max_{z' \in \mathbb{B}_\infty(z, \epsilon)} h^{(y')}_\psi(z') - h^{(y)}_\psi(z').$$

If the solution of the above optimization problem is less than zero for each $y' \neq y$, then robustness of the classifier $h_\psi$ is provably established. Note that, the data consumer can employ same methods as the data producer to encode the classifier as MILP [16] and benefit from any corresponding advancements in solving MILP instances in the context of neural network certification, e.g., [54].

We now formalize our certificate, that allows the data consumer to prove individual fairness of $M$, once given $z$ and $\epsilon$ by the data producer:

**Theorem 1.** *(Individual fairness certificate) Suppose $M = h_\psi \circ f_\theta$ with data point $x$ and similarity notion $\phi$. Furthermore, let $z = f_\theta(x)$, $S_\phi(x) = \{x' \in \mathbb{R}^n \mid \phi(x, x')\}$ and $\epsilon = \max_{x' \in S_\phi(x)} ||z - f_\theta(x')||_\infty$. If*

$$\max_{z' \in \mathbb{B}_\infty(z, \epsilon)} h^{(y')}_\psi(z') - h^{(y)}_\psi(z') < 0$$

*for all labels $y'$ different from the true label $y$, then for all $x' \in S_\phi(x)$ we have $M(x) = M(x')$.*

*Proof.* Provided in Appendix E. $\square$

Table 1: Accuracy and certified individual fairness. We compare the accuracy and percentage of certified individuals with a baseline obtained from setting the loss balancing factor $\gamma = 0$. LCIFR produces a drastic increase in certified individuals while only incurring minor decrease in accuracy.

| CONSTRAINT | DATASET | ACCURACY (%) | | CERTIFIED (%) | |
|---|---|---|---|---|---|
| | | BASE | LCIFR | BASE | LCIFR |
| NOISE | ADULT | 83.0 | 81.4 | 59.0 | 97.8 |
| | COMPAS | 65.8 | 63.4 | 32.1 | 79.0 |
| | CRIME | 84.4 | 83.1 | 7.4 | 66.9 |
| | GERMAN | 76.5 | 74.0 | 71.0 | 97.5 |
| | HEALTH | 80.8 | 81.1 | 75.4 | 97.8 |
| | LAW SCHOOL | 84.4 | 84.6 | 57.9 | 69.2 |
| CAT | ADULT | 83.3 | 83.1 | 79.9 | 100 |
| | COMPAS | 65.6 | 66.3 | 90.9 | 100 |
| | CRIME | 84.4 | 83.9 | 78.3 | 100 |
| | GERMAN | 76.0 | 75.5 | 88.5 | 100 |
| | HEALTH | 80.7 | 80.9 | 64.1 | 99.8 |
| | LAW SCHOOL | 84.4 | 84.4 | 25.6 | 51.1 |
| CAT + NOISE | ADULT | 83.3 | 81.3 | 47.5 | 97.6 |
| | COMPAS | 65.6 | 63.7 | 30.9 | 75.6 |
| | CRIME | 84.4 | 81.5 | 6.2 | 63.3 |
| | GERMAN | 76.0 | 70.0 | 68.0 | 95.5 |
| | HEALTH | 80.7 | 80.7 | 24.7 | 97.3 |
| | LAW SCHOOL | 84.4 | 84.5 | 11.6 | 28.9 |
| ATTRIBUTE | ADULT | 83.0 | 80.9 | 49.3 | 94.6 |
| | GERMAN | 76.5 | 73.5 | 65.0 | 96.5 |
| | LAW SCHOOL | 84.3 | 86.9 | 46.4 | 62.6 |
| QUANTILES | LAW SCHOOL | 84.2 | 84.2 | 56.5 | 76.9 |

# 6 Experimental Evaluation

We implement our method in a tool called LCIFR and present an extensive experimental evaluation. We consider a variety of different datasets — Adult [55], Compas [56], Crime [55], German [55], Health (`https://www.kaggle.com/c/hhp`), and Law School [57] — which we describe in detail in Appendix F. We perform the following preprocessing on all datasets: (i) normalize numerical attributes to zero mean and unit variance, (ii) one-hot encode categorical features, (iii) drop rows and columns with missing values, and (iv) split into train, test and validation sets. Although we only consider datasets with binary classification tasks, we note that our method straightforwardly extends to the multiclass case. We perform all experiments on a desktop PC using a single GeForce RTX 2080 Ti GPU and 16-core Intel(R) Core(TM) i9-9900K CPU @ 3.60GHz. We make all code, datasets and preprocessing pipelines publicly available at `https://github.com/eth-sri/lcifr` to ensure reproducibility of our results.

**Experiment setup** We model the encoder $f_\theta$ as a neural network, and we use logistic regression as a classifier $h_\psi$. We perform a grid search over model architectures and loss balancing factors $\gamma$ which we evaluate on the validation set. As a result, we consider $f_\theta$ with 1 hidden layer of 20 neurons (except for Law School where we do not have a hidden layer) and a latent space of dimension 20. We fix $\gamma$ to 10 for Adult, Crime, and German, to 1 for Compas and Health, and to 0.1 for Law School. We provide a more detailed overview of the model architectures and hyperparameters in Appendix G.

**Fairness constraints** We propose a range of different constraints for which we apply our method. These constraints define the similarity between two individuals based on their numerical attributes (NOISE), categorical attributes (CAT), or combinations thereof (CAT + NOISE). Furthermore, we consider more involved similarity notions based on disjunctions (ATTRIBUTE) and quantiles of certain attributes to counter subordination between social groups [18] (QUANTILES). A full formalization of our constraints is found in Appendix H.

Table 2: Accuracy and percentage of certified individuals for transferable representation learning on Health dataset with CAT + NOISE constraint. The transfer labels are omitted during training and the data producer objective is augmented with a reconstruction loss. This allows the data consumer to achieve high accuracies and certification rates across a variety of (potentially unknown) tasks.

| TASK | LABEL | ACCURACY (%) | CERTIFIED (%) |
|---|---|---|---|
| ORIGINAL | CHARLSON INDEX | 73.8 | 96.9 |
| | MSC2A3 | 73.7 | 86.1 |
| | METAB3 | 75.4 | 93.6 |
| TRANSFER | ARTHSPIN | 75.4 | 93.7 |
| | NEUMENT | 73.8 | 97.1 |
| | RESPR4 | 72.4 | 98.4 |

**Applying our method in practice**  We assume that the data regulator has defined the above constraints. First, we act as the data producer and learn a representation that enforces the individual fairness constraints using our method from Section 4. After training, we compute $\epsilon$ for every individual data point in the test set and pass it to the data consumer along with the latent representation of the entire dataset as described in Section 5.1. Second, we act as data consumer and use our method from Section 4 to learn a locally-robust classifier from the latent representation. Finally, to obtain a certificate of individual fairness, we use $\epsilon$ to certify the classifier via our method from Section 5.2.

In Table 1 we compare the accuracy and percentage of certified individuals (i.e., the empirical approximation of a lower bound on Equation (1)) with a baseline encoder and classifier obtained from standard representation learning (i.e., $\gamma = 0$). We do not compare with other approaches for learning individually fair representations since they either consider a different similarity metric or employ nonlinear components that cannot be efficiently certified. It can be observed that LCIFR drastically increases the percentage of certified individuals across all constraints and datasets. We would like to highlight the relatively low (albeit still significantly higher than baseline) certification rate for the Law School dataset. This is due to the relatively small loss balancing factor $\gamma = 0.1$ which only weakly enforces the individual fairness constraint during training. Finally, we report the following mean certification runtime per input, averaged over all constraints: 0.29s on Adult, 0.35s on Compas, 1.23s on Crime, 0.28s on German, 0.68s on Health, and 0.02s on Law School, showing that our method is computationally efficient. We show that our method scales to larger networks in Appendix I.

**Fair Transfer Learning**  We follow Madras et al. [9] to demonstrate that our method is compatible with transferable representation learning. We also consider the Health dataset, for which the original task is to predict the Charlson Index. To demonstrate transferability, we omit the primary condition group labels from the set of features, and try to predict them from the latent representation without explicitly optimizing for the task. To that end, the data producer additionally learns a decoder $g(z)$, which tries to predict the original attributes $x$ from the latent representation, thereby not only retaining task-specific information on the Charlson Index. This amounts to adding a reconstruction loss $\mathcal{L}_R(x, g(f_\theta(x)))$ (e.g., $\ell_2$) to the objective in Equation (5). Assuming that our representations are in fact transferable, the data consumer is now free to choose any classification objective. We note that our certification method straightforwardly extends to all possible prediction tasks allowing the data consumer to obtain fairness certificates regardless of the objective. Here, we let the data consumer train classifiers for both the original task and to predict the 5 most common primary condition group labels. We display the accuracy and percentage of certified data points on all tasks in Table 2. The table shows that our learned representation transfers well across tasks while additionally providing provable individual fairness guarantees.

## 7  Conclusion

We introduced a novel end-to-end framework for learning representations with provable certificates of individual fairness. We demonstrated that our method is compatible with existing notions of fairness, such as transfer learning. Our evaluation across different datasets and fairness constraints demonstrates the practical effectiveness of our method.

## Broader Impact

Methods that learn from data can potentially produce unfair outcomes by reinforcing human biases or discriminating amongst specific groups. We illustrate how our method can be employed to address these issues with an example due to Cisse and Koyejo [7]. Consider a company with several teams working with the same data to build models. Although the individual teams may not care about fairness of their models, the company needs to comply with ethical or legal requirements. In this setting, our framework enables the company to obtain such certificates from every team in a minimally invasive and modular fashion without compromising downstream utility.

Although individual fairness is a desirable property, it is far from sufficient to provide any ethical guarantees. For example, treating all individuals similarly badly does not conflict with individual fairness. Our method thus depends on the assumption that all involved parties act reasonably. That is, the data regulator needs to take all ethical aspects and future societal consequences into consideration when designing the similarity property. However, even a diligent data regulator may unconsciously encode biases in the similarity measure. Moreover, our approach breaks down with an adversarial data producer that either explicitly learns a discriminatory representation or simply fails to respect the defined similarity notion. Finally, the case where the data consumer acts adversarially has been investigated in previous work [9] and can be mitigated to some extent.

## Acknowledgments and Disclosure of Funding

We would like to thank Anna Chmurovič for her help with initial investigations on combining fairness and differentiable logic during her ETH Student Summer Research Fellowship. We also thank the anonymous reviewers for their insightful comments and suggestions. None of the authors received third party funding or third party support to pursue this work during the last 36 months prior to submission.

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
