[Supplementary Material]

Table 3: Accuracy and certified individual fairness for the CAT + NOISE constraint on the CRIME dataset for different loss balancing factors $\gamma$. Compared to the baseline $\gamma = 0$, our method ($\gamma \neq 0$) incurs minimal changes in accuracy while significantly increasing the percentage of certified individual fairness for a wide range of $\gamma$.

| $\gamma$ | 0 | 0.1 | 0.2 | 0.5 | 1 | 2 | 5 | 10 | 20 | 50 |
|---|---|---|---|---|---|---|---|---|---|---|
| ACCURACY (%) | 84.36 | 84.62 | 84.87 | 84.36 | 84.10 | 84.62 | 84.36 | 81.79 | 50.77 | 50.77 |
| CERTIFIED (%) | 6.15 | 9.23 | 12.05 | 18.46 | 33.08 | 52.31 | 61.28 | 62.82 | 100 | 100 |

## A  Training with Categorical Constraints

A key challenge arising in Section 4 is that DL2 does not support logical formulas $\phi$ involving categorical constraints, which are critical to the fairness context. To illustrate this problem, we recall the example similarity constraint

$$\phi(x, x') := \bigwedge_{i \in \text{Cat} \backslash \{\text{race}\}} (x_i = x'_i) \bigwedge_{j \in \text{Num}} |x_j - x'_j| \leq \alpha.$$

As mentioned in Section 4, numerical attribute constraints of the form $|x_j - x'_j| \leq \alpha$ can be solved efficiently by projecting $x'_j$ onto $[x_j - \alpha, x_j + \alpha]$. Unfortunately, this does not directly extend to categorical constraints. To see this, consider another constraint that considers two individuals $x$ and $x'$ similar irrespective of their race. Further, consider an individual $x$ with only one (categorical) attribute, namely $x = [\text{race}_1]$, and $r$ distinct races. After a one-hot encoding, the features of $x$ are $[1, 0, \ldots, 0]$. Now, one could try to translate the constraint as $|x_k - x'_k| \leq \alpha$ for all $k = 1, \ldots, r$. However, choosing e.g., $\alpha = 0.3$ would only allow for $x'$ with features of the form $[0.7, 0.3, 0, \ldots, 0]$ which still represent the same race when considering the maximum element. Thus, this translation would not consider individuals with different races as similar. At the same time, choosing a larger $\alpha$, e.g., $\alpha = 0.9$, would yield a translation which considers an individual $x'$ with features $[0.9, 0.9, \ldots, 0.9]$ similar to $x$. Clearly, this does not provide a meaningful relaxation of the categorical constraint.

To overcome this problem, we relax the categorical constraint to $x'_k \in [0, 1]$ and normalize the sum over all possible races as $\sum_k x'_k = 1$ with every projection step, thus ensuring a meaningful feature vector. Moreover, it can be easily seen that our translation allows $x'$ to take on any race value irrespective of the race of $x$. We note that although our relaxation can produce features with fractional values, e.g., $[0, 0.2, 0.3, 0, \ldots, 0.5]$, we found it works well in practice.

## B  Loss Balancing Factor $\gamma$

Here, we investigate the impact of the balance parameter $\gamma$ from Equation (5) on the accuracy-fairness tradeoff. To that end, we compare accuracy and certified individual fairness for different loss balancing factors for the CAT + NOISE constraint on the CRIME dataset in Table 3. We observe that increasing $\gamma$ up to 10 yields significant fairness gains while keeping the accuracy roughly constant. For larger values of $\gamma$, the fairness constraint dominates the loss and causes the classifier to resort to majority class prediction, which is perfectly fair. Note that our method can increase both accuracy, albeit only by a small amount, and fairness for certain values of $\gamma$ (e.g., $\gamma = 2$). We conjecture that this effect is due to randomness in the training procedure and sufficient model capacity for simultaneous accuracy and fairness for $\gamma \leq 5$. As we observed the same trend on all datasets, we recommend data producers who want to apply LCIFR in practice to increase $\gamma$ up to the point where the downstream validation accuracy drops below their requirements.

## C  Robust Training

Here, we investigate the necessity and impact of the robust training employed by the data consumer as outlined in Section 4. We recall that the data consumer obtains the latent representation $z = f_\theta(x)$ for every data point $x$ from the data producer. Assuming that the latent representation was generated by an encoder $f_\theta$ trained to maintain predictive utility and satisfy Equation (4), the data consumer

only needs to ensure local robustness of her classifier $h_\psi$ for perturbations up to $\delta$ in $\ell_\infty$-norm to obtain an individually fair classifier $h_\psi$. However, the data consumer may be hestitant to apply robust training methods due to potentially negative impacts on accuracy or may not care about fairness at all.

We first study the case where the data consumer employs logistic regression for $h_\psi$ as in Section 6. We consider the CAT + NOISE constraint and run LCIFR on all five datasets for different loss balancing factors $\gamma \in \{0.001, 0.01, 0.1, 1, 10\}$ both with and without adversarial training for $h_\psi$. Across all datasets and values of $\gamma$ the largest increase in certification for adversarial training is roughly 7%, with a simultaneous accuracy drop of 0.5%, and the largest accuracy drop is roughly 1%, with a simultaneous increase in certification of 2.9%. This rather limited impact of adversarial training on both accuracy and certifiable individual fairness for logistic regression is to be expected due to the smoothness of the decision boundary. However, for a more complex classifier, such as a feedforward neural network with 2 hidden layers of 20 nodes each, adversarial training doubles the certification rate from 34% to 70.8%, while decreasing accuracy only by 1.6% for the CAT + NOISE constraint on the HEALTH datasets with $\gamma = 1$.

## D  Full Encoding

Here, we present our fairness constraint language and show how to encode constraints as a mixed-inter linear program (MILP). We closely follow Fischer et al. [15].

**Logical language**   We recall that our framework allows the data regulator to define notions of similarity via a logical constraint $\phi$. Our language of logical constraints consists of boolean combinations of comparisons between terms where each term $t$ is a linear function over a data point $x$. We note that although Fischer et al. [15] support terms with real-valued functions, we only consider linear functions since nonlinear constraints, e.g., $x^2 < 3$, cannot be encoded exactly as MILP. Unlike Fischer et al. [15], our constraint language also supports constraints on categorical features. To form comparison constraints, two terms $t$ and $t'$ can be combined as $t = t'$, $t \leq t'$, $t \neq t'$, and $t < t'$. Finally, a logical constraint $\phi$ is either a comparison constraint, a negation $\neg\phi'$ of a constraint $\phi'$, or a conjunction $\phi' \wedge \phi''$ or disjunction $\phi' \vee \phi''$ of two constraints $\phi'$ and $\phi''$.

**Encoding as MILP**   Given an individual $x$ and a logical constraint $\phi$ capturing some notion of similarity, the data producer needs to compute the radius $\epsilon$ of the smallest $\ell_\infty$-ball around the latent representation $z = f_\theta(x)$ that contains the latent representations of all similar individuals $f_\theta(S_\phi(x))$, i.e., $\arg\min_\epsilon f_\theta(S_\phi(x)) \subseteq \mathcal{B}_\infty(z, \epsilon)$. To that end, the data producer is required to encode $S_\phi(x)$ as a MILP which can be performed in a recursive manner.

The individual $x$ belongs to the test dataset and can thus be treated as a constant. To model $S_\phi(x)$, we encode a similar individual $x'$ by considering numerical and categorical features separately. For all numerical features we add a real-valued variable $v_i$ to the MILP. For all categorical features we add $k_j$ binary variables $v_j^l$ for $l = 1, \ldots, k_j$, where $k_j$ is the number of distinct values this categorical feature can take, to the MILP. Furthermore, we add the constraint $\sum_l v_j^l = 1$ for every categorical variable, thereby ensuring that it takes on one and only one of its values.

With these variables, each term can be directly encoded as it consists of a linear function. Likewise, the comparison constraints $=$, $\leq$, and $<$ can be directly encoded in the MILP. We encode $t \neq t'$ as $(t < t') \vee (t' < t)$ for continuous variables and as $\bigvee_{l \neq t'} t = l$ for categorical variables.

Next, we consider the case where $\phi$ is a boolean combination of constraints $\phi' \wedge \phi''$ or $\phi' \vee \phi''$. The first case can be encoded straightforwardly in the MILP. To encode the disjunction $\phi' \vee \phi''$ we add two additional binary variables $v'$ and $v''$ to the MILP with the constraints

$$
\begin{aligned}
v' = 1 &\iff \phi', \\
v'' = 1 &\iff \phi'', \\
v' + v'' &\geq 1.
\end{aligned}
$$

Finally, if $\phi$ is a negation $\neg\phi'$ of $\phi'$, the constraint is preprocessed and rewritten into a logically equivalent constraint before encoding as MILP:

$$\neg\,(t = t') := t \neq t',$$
$$\neg\,(t \leq t') := t' < t,$$
$$\neg\,(t \neq t') := t = t',$$
$$\neg\,(t < t') := t' \leq t,$$
$$\neg\,(\phi' \wedge \phi'') := \neg\phi' \vee \neg\phi'',$$
$$\neg\,(\phi' \vee \phi'') := \neg\phi' \wedge \neg\phi'',$$
$$\neg\,(\neg\phi') := \phi'.$$

# E    Individual Fairness Certificate

In this section, we prove the correctness of our individual fairness certificate as formalized in Theorem 2, which allows the data consumer to prove individual fairness of the end-to-end model $M$, once given the latent representation $z$ and radius $\epsilon$ by the data producer:

**Theorem 2.** *(Individual fairness certificate) Suppose $M = h_\psi \circ f_\theta$ with data point $x$ and similarity notion $\phi$. Furthermore, let $z = f_\theta(x)$, $S_\phi(x) = \{x' \in \mathbb{R}^n \mid \phi(x, x')\}$ and $\epsilon = \max_{x' \in S_\phi(x)} ||z - f_\theta(x')||_\infty$. If*

$$\max_{z' \in \mathbb{B}_\infty(z,\epsilon)} h_\psi^{(y')}(z') - h_\psi^{(y)}(z') < 0$$

*for all labels $y'$ different from the true label $y$, then for all $x' \in S_\phi(x)$ we have $M(x) = M(x')$.*

*Proof.* The data producer computes the latent representation $z = f_\theta(x)$ and certifies that

$$\epsilon = \max_{x' \in S_\phi(x)} ||z - f_\theta(x')||_\infty. \tag{6}$$

Thus, it immediately follows that $f_\theta(S_\phi(x)) \subseteq \mathbb{B}_\infty(z, \epsilon)$, where $\mathbb{B}_\infty(z, \epsilon)$ is the $\ell_\infty$-bounding box with center $z$ and radius $\epsilon$. Consider any label $y'$ different from the true label $y = M(x)$. If the data consumer certifies that

$$\max_{z' \in \mathbb{B}_\infty(z,\epsilon)} h_\psi^{(y')}(z') - h_\psi^{(y)}(z') < 0, \tag{7}$$

then the classifier will predict label $y$ for all $z' \in \mathbb{B}_\infty(z, \epsilon)$. Combining this with $f_\theta(S_\phi(x)) \subseteq \mathbb{B}_\infty(z, \epsilon)$ we have

$$\forall x' \in S_\phi(x) : M(x) = M(x'),$$

implying that the end-to-end classifier is individually fair for similarity notion $\phi$ at data point $x$. We refer to Section 5 and Tjeng et al. [16] for details on the correctness of the certificates for Equations (6) and (7).                                                                                                                             $\square$

# F    Datasets

In this section, we provide a detailed overview of the datasets considered in Section 6. We recall that we perform the following preprocessing on all datasets: (i) normalize numerical attributes to zero mean and unit variance, (ii) one-hot encode categorical features, (iii) drop rows and columns with missing values, and (iv) split into train, test and validation sets. Although we only consider datasets with binary classification tasks, we note that our method straightforwardly extends to the multiclass case.

**Adult**    The Adult Income dataset [55] is extracted from the 1994 US Census database. Every sample represents an individual and the goal is to predict whether that person's income is over 50K$ / year.

**Compas**    The COMPAS Recidivism Risk Score dataset contains data collected on the use of the COMPAS risk assessment tool in Broward County, Florida Angwin [56]. The task is to predict recidivism within two years for all individuals.

Table 4: Statistics for train, validation, and test datasets. Note that most of the datasets, namely Adult, German, Health, and Law School, have a highly skewed distribution of positive labels.

| | TRAIN | | VALIDATION | | TEST | |
|---|---|---|---|---|---|---|
| | SIZE | POSITIVE | SIZE | POSITIVE | SIZE | POSITIVE |
| ADULT | 24129 | 24.9% | 6033 | 24.9% | 15060 | 24.6% |
| COMPAS | 3377 | 52.3% | 845 | 52.2% | 1056 | 55.6% |
| CRIME | 1276 | 48.7% | 319 | 55.5% | 399 | 49.6% |
| GERMAN | 640 | 70.5% | 160 | 66.9% | 200 | 71.0% |
| HEALTH | 139785 | 68.0% | 34947 | 68.6% | 43683 | 68.0% |
| LAW SCHOOL | 5053 | 27.3% | 13764 | 26.8% | 17205 | 26.3% |

Table 5: Percentage of positive labels for train, validation, and test datasets for transfer learning tasks. Note, that the percentages do not sum to 100% as the labels are aggregated by patient and year.

| | POSITIVE (%) | | |
|---|---|---|---|
| | TRAIN | VALIDATION | TEST |
| MSC2A3 | 62.0 | 61.9 | 61.9 |
| METAB3 | 34.9 | 34.9 | 34.9 |
| ARTHSPIN | 31.5 | 31.7 | 32.1 |
| NEUMENT | 28.4 | 28.5 | 28.6 |
| RESPR4 | 27.5 | 27.5 | 27.5 |

**Crime** The Communities and Crime dataset [55] contains socio-economic, law-enforcement, and crime data for communities within the US. We try to predict whether a specific community is above or below the median number of violent crimes per population.

**German** The German Credit dataset [55] contains 1000 instances describing individuals who are either classified as good or bad credit risks.

**Health** The Heritage Health dataset (`https://www.kaggle.com/c/hhp`) contains physician records and insurance claims. For every patient we try to predict ten-year mortality by binarizing the Charlson Index, taking the median value as a cutoff.

**Law School** This dataset from the Law School Admission Council's National Longitudinal Bar Passage Study [57] has application records for 25 different law schools. The task is to predict whether a student passes the bar exam.

We note that for some of these datasets the label distribution is highly unbalanced as displayed in Table 4. For example, for the Law School dataset, learning a representation that maps all individuals to the same point in the latent space and classifying that point as negative would yield 73.7% test set accuracy. Moreover, individual fairness would be trivially satisfied for any constraint $\phi$ as all individuals are mapped to the same outcome. It is thus important to compare the performance of all models with the base rates from Table 4. Moreover, for every table containing accuracy values we provide an analogous table with balanced accuracy in Appendix J.

**Fair Transfer Learning** We follow Madras et al. [9] and consider the Health dataset for transferable representation learning. The original task for the Health dataset is to predict the Charlson Index. Thus, to demonstrate transferability, we omit the primary condition group labels from the set of features, and try to predict them from the latent representation without explicitly optimizing for the task. We display the (highly imbalanced) label distributions for the considered primary condition groups in Table 5.

# G Experiment Setup

Here, we provide a detailed overview of the model architectures and training hyperparameters considered in Section 6. Recall that we model the encoder $f_\theta$ as a neural network, and we use logistic regression as a classifier $h_\psi$. We run a grid search over model architectures and loss balancing factors $\gamma$ which we evaluate on the validation set. Concretely, we search over two different encoders (both with latent space of dimension 20): (i) without a hidden layer and (ii) with a single hidden layer of 20 neurons, and loss balancing factors $\gamma \in [10, 1, 0, 0.01]$. As a result, we consider $f_\theta$ with one hidden layer of 20 neurons (except for Law School where we do not have a hidden layer) and a latent space of dimension 20. We fix $\gamma$ to 10 for Adult, Crime, and German, to 1 for Compas and Health, and to 0.1 for Law School. We train our models for 100 epochs with a batch size of 256. We use the Adam optimizer [51] with weight decay 0.01 and dynamic learning rate scheduling based on validation measurements (ReduceLROnPlateau from [58]) starting at 0.01 with a patience 5 of epochs. Finally, we run DL2 with 25 PGD iterations with step size 0.05 to find counterexamples (cf. Section 4).

# H Constraints

In this section, we provide a full formalization of the similarity constraints considered in Section 6.

**Noise (NOISE)**   Under this constraint, two individuals are similar if their normalized numerical features differ by no more than $\alpha$. We consider $\alpha = 0.3$ for all experiments, which means e.g., for Adult: two individuals are similar if their age difference is smaller than roughly 3.95 years.

**Categorical (CAT)**   We consider two individuals similar if they are identical except for one or multiple categorical attributes. For Adult and German, we choose the binary attribute gender. For Compas, two people are to be treated similarly regardless of race. For Crime, we enforce the constraint that the state should not affect prediction outcome for two neighborhoods. For Health, two identical patients, except for gender and age, should observe the same ten-year mortality at their first insurance claim. For Law School, we consider two individuals similar regardless of their race and gender.

**Categorical and noise (CAT + NOISE)**   This constraint combines the two previous constraints and considers two individuals as similar if their numerical features differ no more than $\alpha$ regardless of their values for certain categorical attributes.

**Conditional attributes (ATTRIBUTE)**   In this case, $\phi$ is composed of a disjunction of two mutually exclusive cases, one of which has to hold for similarity. For this, we consider a numerical attribute and a threshold $\tau$. If two individuals are both below $\tau$, then they are similar if their normalized attribute differences are less than $\alpha_1$. If both individuals are above $\tau$, similarity holds if the attribute differences are less than $\alpha_2$. Concretely, consider two applicants from the Law School dataset. If both of their GPAs are below $\tau = 3.4$ (the median), then they are similar only if their difference in GPA is less than 0.1694 ($\alpha_1 = 0.4$). However, if both their GPAs are above 3.4, then we consider the applicants similar if their GPAs differ less than 0.847 ($\alpha_2 = 0.2$). For Adult, we consider the median age as threshold $\tau = 37$, with $\alpha_1 = 0.2$ and $\alpha_2 = 0.4$ which corresponds to age differences of 2.63 and 5.26 years respectively. For German, we also consider the median age as threshold $\tau = 33$, with $\alpha_1 = 0.2$ and $\alpha_2 = 0.4$ which corresponds to age differences of roughly 0.24 and 0.47 years respectively.

**Subordination (QUANTILES)**   We follow Lahoti et al. [18] and define a constraint that counters subordination between social groups. We consider the Law School dataset and differentiate two social groups by race, one group containing individuals of white race and the other containing all remaining races. To counter subordination, we compute within-group ranks based on the GPAs and define similarity if the rank difference for two students from different groups is less than 24. Thus, two students are considered similar if their performance relative to their group is similar even though their GPAs may differ significantly.

Table 6: Balanced accuracy for encoders and classifiers from Table 1.

| CONSTRAINT | DATASET | BALANCED ACCURACY (%) | |
|---|---|---|---|
| | | BASE | LCIFR |
| NOISE | ADULT | 74.5 | 70.9 |
| | COMPAS | 65.1 | 62.3 |
| | CRIME | 84.4 | 83.2 |
| | GERMAN | 69.6 | 60.8 |
| | HEALTH | 77.1 | 76.5 |
| | LAW SCHOOL | 76.1 | 75.8 |
| CAT | ADULT | 74.7 | 73.9 |
| | COMPAS | 64.9 | 65.7 |
| | CRIME | 84.4 | 83.9 |
| | GERMAN | 69.2 | 68.3 |
| | HEALTH | 77.2 | 77.1 |
| | LAW SCHOOL | 76.1 | 75.5 |
| CAT + NOISE | ADULT | 74.7 | 70.8 |
| | COMPAS | 64.9 | 62.5 |
| | CRIME | 84.4 | 81.7 |
| | GERMAN | 69.2 | 49.8 |
| | HEALTH | 77.2 | 76.5 |
| | LAW SCHOOL | 76.1 | 75.5 |
| ATTRIBUTE | ADULT | 74.5 | 70.1 |
| | GERMAN | 69.6 | 61.8 |
| | LAW SCHOOL | 76.1 | 74.3 |
| QUANTILES | LAW SCHOOL | 76.1 | 75.8 |

## I  Scaling to Large Networks

To show that our method can be easily scaled to larger networks, we train an encoder $f_\theta$ with 200 hidden neurons and latent space dimension 200. For such large models we can relax the MILP encodings to a linear program [59] and solve for robustness via convex relaxation. Running this relaxation for our large network and the NOISE constraint on Adult we can certify fairness for 91.4% of the individuals with 82.8% accuracy and average certification runtime of 1.13s. In contrast, the complete solver can certify 92.6% of individuals with average runtime of 31.9s. For even larger model architectures, one can use one of the recent state-of-the-art network verifiers [46].

## J  Balanced Accuracy

We recall that some of the datasets are highly imbalanced (cf. Table 4). Hence, we evaluate the balanced accuracies for the models from Table 1 and show them in Table 6. It can be observed that LCIFR performs only slightly worse than the baseline across all constraints and datasets (except for CAT + NOISE on German).

**Fair Transfer Learning**  We recall that the label distribution of the primary condition groups (transfer tasks) are highly imbalanced (cf. Table 5). Nevertheless, LCIFR achieves accuracies that are above the base rate achieved by majority class prediction (cf. Table 1) in all cases except for RESPR4. Here, we display the corresponding balanced accuracies in Table 7, and we observe that the balanced accuracies are inversely proportional to the label imbalance (cf. Table 5).

Table 7: Balanced accuracy for transferable representation learning on Health dataset with CAT + NOISE constraint from Table 2.

| TASK | LABEL | BALANCED ACCURACY (%) |
|------|-------|------------------------|
| ORIGINAL | CHARLSON INDEX | 63.9 |
| TRANSFER | MSC2A3 | 70.8 |
| | METAB3 | 68.5 |
| | ARTHSPIN | 66.0 |
| | NEUMENT | 58.9 |
| | RESPR4 | 56.0 |