[Reviews · NeurIPS 2020]

Review 1

Summary and Contributions: This paper offers a method to train individually fair representations. The process follows as (1) a data regulator defines a set of similarity notions (e.g. individuals are similar if x categorical features are the same and y numerical features are leq some difference). (2) A data producer trains a representation where individuals considered similar are mapped within a specified distance within the latent space. To perform training, they use a method based on training neural networks using logical constraints called DL2 --- here, the logical constraints are given by the data regulator. (3) A data consumer trains a model on the representation. Here the data consumer must ensure local robustness of the classifier in order for it to be individually fair. While the representation maps similar individuals "close together" in the latent space, an arbitrary classifier may not be individually fair, so the data consumer must ensure that the trained classifier is locally robust. The authors provide comprehensive evaluation on a variety of data sets and problem settings. They find that their model produces classifiers that have high degrees of individual fairness.

Strengths: This paper presents an interesting and novel method to train individually fair representations. Representational fairness has been of interest at NeurIPS, so this work is relevant to the NeurIPS community. Training the fair representation through logical constraints imposed by the data regulator is a particularly compelling, as it presents a route for non or semi technical data regulators to articulate key individual fairness constraints. This method is significant because it could open up a number of new approaches to representational fairness because of the intuitiveness of providing logical constraints. Further, the empirical results are convincing. The method offers minimal changes to accuracy and produces significant changes to individual fairness. Last, its worth noting that the computational time imposed by the method is minimal, which is positive.

Weaknesses: One of the advantages of group fair representation learning is that the data consumer doesn't have to train the classifier in any particular fashion. Here, the data consumer needs to be fairness aware and capable of training the classifier to satisfy the needed local robustness property. In this way, the method isn't as widely applicable as related group fair representation learning strategies. Still, the work provides a good start towards training individually fair representations. Empirically, it would be interesting to see the accuracy / individual fairness of the logistic regression trained on the representation, without training for local robustness. While the theoretical necessity is clear, this would help readers understand the necessity of this step empirically.

Correctness: Claims and methodology seem correct.

Clarity: This paper is well written and easy to read.

Relation to Prior Work: Yes, prior work is well situated.

Reproducibility: Yes

Additional Feedback: Update: Thanks for the responses.


Review 2

Summary and Contributions: The paper presents an end-to-end framework for learning representations with certified individual fairness. The paper follows the framework setting in [10] that contains a data regulator, a data producer, and a data consumer. The data producer and the data consumer are independent. The main contributions of this paper include: 1) individual fairness notions defined via interpretable logical constraints, 2) the first use of training ML models with logical constraints [15] and the constraint satisfaction checking method [16] for learning certified individually fair models, and 3) an open source tool together with extensive empirical evaluation.

Strengths: + The introduced individual fairness notions based on logical constraints are new and can be adopted in many applications as they can provide interpretable and domain-specific fair notions. + The proposed framework was built upon several newly developed techniques, training ML models with logical constraints via a translation from logical constraints to a differentiable loss function [15], new methods for proving that constraints are satisfied, and modeling the encoder as a mixed-integer linear program optimization problem. While each of the above techniques cannot be considered as this paper's contribution, putting them together for learning certified individually fair representations and demonstrating feasibility and effectiveness are indeed this paper's contribution. + The opensource tool together with the well studied empirical evaluation are another strength.

Weaknesses: I do not identify any significant weakness of this work. Some minor comments are below. Some discussions about comparisons with counterfactual individual fairness and indirect fairness (via redlining attributes) would be helpful for readers to better understand the applicability of the proposed notions. The balance parameter \gamma is important. While the paper specified different \gamma for different datasets and compare them with the baseline with \gamma =0, it would be better to pick one dataset (e.g., law school) with varying \gamma values. Then readers can better know the accuracy-fairness tradeoff with different balance values.

Correctness: The claims and the developed methods are correct. The empirical methodology is also correct.

Clarity: The paper is very well written and can be easily followed. This paper is also a good mix of theoretical analysis and empirical evaluation.

Relation to Prior Work: The claimed contributions and descriptions of relation to prior work are generally clear. The comparisons with [10] from setting and approach perspectives could be more clear. I even wondered whether empirical comparisons with [10] should be conducted.

Reproducibility: Yes

Additional Feedback: The authors may consider to move fair transfer learning from Appendix G to the main body as this could be another major contribution. Thanks for authors' rebuttal feedback, in particular, on comparisons with counterfactual individual fairness and indirect fairness, and the balance parameter \gamma.


Review 3

Summary and Contributions: The paper introduces a framework for learning representations of data with a certified individual fairness guarantee. Such framework is composed of two independent components that operate in sequence: (1) a data producer who learns an encoder that maps the data into a latent space, with the goal that similar individuals after transformation are still close to each other; and (2) a data consumer who learns a certified robust classifier on the transformed data. The composition of two ensures certified individual fairness. Both the encoder and the robust classifier can be formulated as min-max optimization and can be solved efficiently using the existing algorithms.

Strengths: 1. Learning data representation with a certified fairness guarantee is an important research topic in machine learning. The fair data representation can be used for various downstream tasks in a variety of applications. 2. The certified individual fairness is achieved in two steps by data producer and data consumer; two components can work independently and are not constrained by each other. The proposed method is NOT domain/application-specific that there is no constraint imposed on the similarity metric in fairness criterion. 3. The framework proposed in the paper is flexible and can be achieved efficiently by utilizing the existing tools and algorithms. For example, the mixed-integer linear program solvers can be used to find convex relaxation of the latent set (radius epsilon); the algorithms in robust machine learning can be used by data producer to certify individual fairness.

Weaknesses: UPDATE: I appreciate the author's clarification on the comparison with local DP, the notations \phi, \mu, and the additional experiments on examining the effect of loss balancing factor. As an additional suggestion, it would be better if authors can include more related work on the topic of learning robust representation, such as "Zhu, Sicheng, Xiao Zhang, and David Evans. Learning Adversarially Robust Representations via Worst-Case Mutual Information Maximization", "Garg, Shivam, et al. A spectral view of adversarially robust features.", "Pensia, Ankit, Varun Jog, and Po-Ling Loh. Extracting robust and accurate features via a robust information bottleneck" and so on. While I believe these studies are different from this paper, I think this topic is highly relevant to the topic of learning fair representation. 1. The performance highly depends on the choice of classifiers/parameters (e.g., loss balancing factors) and datasets, neither theoretical guarantee nor the guidance on the selection of classifiers/parameters is provided, which is one of the main reasons that limit my score. The performance is only compared with the BASE case. I wonder if authors can compare (at least empirically) with other fair representation learning methods. 2. The individual fairness studied in the paper — treating similar people similarly — is highly related to differential privacy and robustness ML. In essence, all are trying to guarantee that any small perturbation of training data cannot change the output significantly. Their relations have been studied extensively in the literature. It is thus NOT surprising to see the use of robust learning in providing certificates for individual fairness. Moreover, most of the methods in the paper are adopted from the existing work (e.g., the use of logical constraints, the formulation of min-max optimization, etc.). I am concerned that the work doesn’t have significant novelty and contribution. It would be helpful if authors can explain how their framework differs fundamentally from the existing work, especially those in the domain of robust and (local) differential privacy. 3. When the fairness constraint is imposed in machine learning, typically there should be a tradeoff between accuracy and fairness. I believe there is also such a tradeoff in the proposed framework, which is an important criterion but has not been discussed in the paper. However, as shown in the experiments, both accuracy and fairness can be improved simultaneously in many cases (e.g., HEALTH dataset). It seems that fairness can be attained "for free" without losing accuracy. I wonder if authors can explain why this can happen. I suggest authors conducting more experiments to examine the impact of loss balancing factor on the performance, which I believe plays a critical role in balancing fairness-accuracy tradeoff.

Correctness: Yes

Clarity: The paper is well-organized and well-written in general. Some parts may be improved. For example, it would be good if the authors can state explicitly with more details in Sec 2 that the similarity measure \phi, \mu are logical expressions taking value either 0 or 1.

Relation to Prior Work: The paper introduces some related work but is not sufficient. As mentioned by authors in related work, some other works have also studied individual fair representation learning. Although those works may focus on the similarity metric that is different from the current paper, their methods should be compared, and the difference should be highlighted. As I mentioned in the "Weakness" section, individual fairness is highly related to differential privacy and robustness, the comparisons with the existing works in these domains are needed.

Reproducibility: Yes

Additional Feedback: In sec 5.2, it seems that $h^{(y)}_{\psi}(z)$ is used directly without definition. How does it differ from $h_{\psi}(z)$? My understanding is that $h^{(y)}_{\psi}(z)$ denotes the likelihood of $z$'s label being predicted as $y$ by classifier $h_{\psi}(z)$, please correct me if I am wrong. Please see "weakness" section for additional suggestions and questions.

[Author Response · NeurIPS 2020]

We thank the reviewers for their insightful feedback and encouraging words. We are pleased that all reviewers
acknowledge the relevance of learning certified individually fair representations. Below, we address the reviewers'
comments and concerns, all of which we will incorporate into the next version of our work.

**R1: Can you investigate the impact of robustly training the classifier on accuracy and certifiability?** The impact
of adversarial training on logistic regression (for both accuracy and fairness) is limited due to the smoothness of the
decision boundary. In fact, over all datasets and a wide range of $\gamma$, the largest increase in certification is roughly 7%,
with a simultaneous accuracy drop of 0.3%. In contrast, for a more complex classifier, such as a feedforward neural
network with 2 hidden layers of 20 nodes each, adversarial training doubles the certification rate (from 34% to 70.8%)
while decreasing the accuracy only by 1.6%. We will provide a more thorough investigation in the next revision.

**R2: How does your work compare with counterfactual and indirect fairness?** In contrast to counterfactual and
indirect fairness, we do not require a causal model of the underlying data distribution. This allows for more flexibility
in applying our approach but prohibits us from making causal/counterfactual claims. Nevertheless, the combination of
logical constraints and causal/counterfactual fairness represents an interesting direction for future research.

**R2: Can you extend your discussion of the framework from McNamara et al. [10]?** Yes, a key difference to [10]
is that their approach requires to know statistics of the data distribution to obtain guarantees. This allows them to
compute probabilistic bounds for individual (and group) fairness: for a new data point from the same distribution, the
constraint will hold with a certain probability. Our result is different in that we obtain a certificate for a fixed data point,
which ensures that the fairness constraint holds (independent of the other data points). While both approaches are valid
and practically relevant, they are also fundamentally different, which renders experimental comparison meaningless.

**R2: You could move fair transfer learning to the main body as this is another major contribution.** We agree that
the compatibility of our method with existing fairness notions is an important contribution, and we would use the
additional space of the camera-ready version to move the fair transfer learning section to the main body.

**R2 & R3: What is the impact of the balance parameter $\gamma$ on the accuracy-fairness tradeoff?** We compare the
accuracy and certified individual fairness for different loss balancing factors $\gamma$ for the CAT + NOISE constraint on the
CRIME dataset in Table 1. We observe that increasing $\gamma$ up to 10 yields significant fairness gains while keeping the
accuracy roughly constant. For larger values of $\gamma$, the fairness constraint dominates the loss and causes the classifier to
resort to majority class prediction (which is perfectly fair). As mentioned by reviewer 3, our method increases both
accuracy (albeit only by a small amount) and fairness for certain values of $\gamma$ (e.g., $\gamma = 2$). Based on our observations, we
conjecture that this effect is due to randomness in the training procedure and sufficient model capacity for simultaneous
accuracy and fairness for $\gamma \leq 5$. As we observed the same trend on all datasets, we recommend data producers to
increase $\gamma$ up to the point where the downstream validation accuracy drops below their requirements.

Table 1: Accuracy and certified individual fairness for the CAT + NOISE constraint on the CRIME dataset for different
loss balancing factors $\gamma$. Compared to the baseline $\gamma = 0$, our method ($\gamma \neq 0$) incurs minimal changes in accuracy
while significantly increasing the percentage of certified individual fairness for a wide range of $\gamma$.

| $\gamma$ | 0 | 0.1 | 0.2 | 0.5 | 1 | 2 | 5 | 10 | 20 | 50 |
|---|---|---|---|---|---|---|---|---|---|---|
| accuracy (%) | 84.36 | 84.62 | 84.87 | 84.36 | 84.10 | 84.62 | 84.36 | 81.79 | 50.77 | 50.77 |
| certified (%) | 6.15 | 9.23 | 12.05 | 18.46 | 33.08 | 52.31 | 61.28 | 62.82 | 100 | 100 |

**R3: Can you compare with other fair representation learning methods?** Yes, we will provide more in-depth
comparison, even though we do not believe works, e.g., Zemel et al. [8], which either do not focus on proximity in latent
space or use nonlinear methods that cannot be efficiently certified, will yield certified individually fair representations.

**R3: Can you comment on the relationship with differential privacy?** The close relationship between individual
fairness and differential privacy (DP) has been discussed in previous work (see, e.g., [12]). However, DP crucially
differs from our work in that it obtains a probabilistic guarantee, similar to [10] mentioned above, whereas we compute
absolute guarantees for every data point. The DP analog of LCIFR is to limit the sensitivity of $f_\theta$ by injecting noise, and
to consider $h_\psi$ as a post-processing step. We will include an extended version of this discussion in the next revision.

**R3: Is your method merely an adoption of existing work?** No, recent advances in training with logical constraints
and proving constraint satisfaction enable us to tackle a previously unsolved problem: provable individual fairness across
*modular* components. As outlined by reviewer 2, learning certified individually fair representations and demonstrating
feasibility and effectiveness are indeed essential contributions.

**R3: Please clarify that $\phi$ and $\mu$ are logical formulas taking value either 0 or 1.** We stated this in L43/L44 and L74.

**R3: How does $h_\psi^{(y)}(z)$ differ from $h_\psi(z)$?** We will clarify that $h_\psi^{(y)}(z)$ is the logit corresponding to label $y$.

[Meta-Review · NeurIPS 2020]

The reviewing committee agrees this is a strong proposal that borrows tools from recent advances in constraint satisfaction to address a critical challenge in ML fairness (certification for individually fair representation). The paper is clearly written and the experiments are well conducted. We recommend its acceptance for publication and suggest that authors incorporate in the final version the clarifications and results included in the rebuttal. The authors are also encourage to include additional baselines in the final version of the paper.